# Genome-Engineered mpkCCDc14 Cells as a New Resource for Studying AQP2

**DOI:** 10.3390/ijms24021684

**Published:** 2023-01-14

**Authors:** Hyo-Ju Jang, Hye-Jeong Park, Hong Seok Choi, Hyun Jun Jung, Tae-Hwan Kwon

**Affiliations:** 1Department of Biochemistry and Cell Biology, School of Medicine, Kyungpook National University, Taegu 41944, Republic of Korea; 2BK21 FOUR KNU Convergence Educational Program, Department of Biomedical Science, School of Medicine, Kyungpook National University, Taegu 41944, Republic of Korea; 3Division of Nephrology, Department of Medicine, Johns Hopkins University School of Medicine, Baltimore, MD 21205, USA

**Keywords:** aquaporin-2, CRISPR/Cas9, genome engineering, vasopressin receptor

## Abstract

mpkCCDc14 cells, a polarized epithelial cell line derived from mouse kidney cortical collecting ducts, are known to express the vasopressin V2 receptor (V2R) and aquaporin-2 (AQP2) that are responsive to vasopressin. However, a low abundance of the endogenous AQP2 protein in the absence of vasopressin and heterogeneity of AQP2 protein abundance among the cultured cells may limit the further application of the cell line in AQP2 studies. To overcome the limitation, we aimed to establish mpkCCDc14 cells constitutively expressing V2R and AQP2 via CRISPR/Cas9-mediated genome engineering technology (i.e., V2R-AQP2 cells). 3′- and 5′-Junction PCR revealed that the V2R-AQP2 expression cassette with a long insert size (~2.2 kb) was correctly integrated. Immunoblotting revealed the expression of products of integrated *Aqp2* genes. Cell proliferation rate and dDAVP-induced cAMP production were not affected by the knock-in of *Avpr2* and *Aqp2* genes. The AQP2 protein abundance was significantly higher in V2R-AQP2 cells compared with control mpkCCDc14 cells in the absence of dDAVP and the integrated AQP2 was detected. Immunocytochemistry demonstrated that V2R-AQP2 cells exhibited more homogenous and prominent AQP2 labeling intensity in the absence of dDAVP stimulation. Moreover, prominent AQP2 immunolabeling (both AQP2 and pS256-AQP2) in the apical domain of the genome-edited cells was observed in response to dDAVP stimulation, similar to that in the unedited control mpkCCDc14 cells. Taken together, mpkCCDc14 cells constitutively expressing V2R and AQP2 via genome engineering could be exploited for AQP2 studies.

## 1. Introduction

Urine concentration and body water homeostasis are controlled by the anti-diuretic hormone, vasopressin, which increases osmotic water permeability and water reabsorption in the kidney collecting duct [1,2]. Aquaporin-2 (AQP2) is the water channel protein localized at the apical plasma membrane and subapical vesicles in the connecting tubule and collecting duct cells, and is regulated by vasopressin on a short-term and a long-term basis for water reabsorption and urine concentration [1,3,4,5,6]. The short-term regulation of AQP2 is dependent on the intracellular translocation of AQP2-expressing vesicles to the apical plasma membrane, resulting in an increase in the osmotic water permeability of the cells [4,7,8,9]. The long-term regulation of AQP2 is based on the changes in AQP2 protein abundance [1,8,10,11]. Vasopressin induces *Aqp2* gene transcription [8,12] and increases the half-life of the AQP2 protein [13]. In contrast, vasopressin withdrawal induces the intracellular degradation of AQP2 protein, mediated by ubiquitination and proteasomal-lysosomal degradation [14,15,16,17].

mpkCCDc14 cells, a polarized epithelial cell line derived from mouse kidney cortical collecting ducts [18], have been used for AQP2 studies since they are known to express both vasopressin V2 receptor (V2R) and AQP2, which are responsive to vasopressin [19,20,21,22]. However, the endogenous AQP2 protein abundance in mpkCCDc14 cells is very low in the absence of vasopressin stimulation. In addition, the heterogeneity of AQP2 protein abundance among the cultured mpkCCDc14 cells was shown in the in vitro culture system [23]. Vasopressin pre-treatment is, therefore, required to induce sufficient AQP2 expression to study the regulation of functional AQP2 proteins in mpkCCDc14 cells. In terms of AQP2 trafficking studies, however, dDAVP pre-treatment may have limitations. Although the induction of endogenous AQP2 expression by the pre-treatment of dDAVP produces detectable endogenous AQP2 proteins, dDAVP withdrawal for 2 or 4 h [24,25] might not fully recover cellular signaling pathways, including kinase activity associated with AQP2 trafficking. Therefore, the pre-treatment of dDAVP for the induction of endogenous AQP2 is likely to cause an artifact in studying cellular processes for AQP2 trafficking. More extended withdrawal may affect total AQP2 abundance, which might not be enough for detection. To overcome these limitations, we aimed: (1) to establish mpkCCDc14 cells stably expressing both V2R and AQP2 via CRISPR/Cas9-mediated genome engineering technology; and (2) to examine whether mpkCCDc14 cells with products from the targeted integration of both *Avpr2* gene and *Aqp2* gene are responsive to 1-deamino-8-D-arginine vasopressin (dDAVP) stimulation, a V2R agonist, to induce intracellular cAMP level and AQP2 translocation into the apical plasma membrane.

## 2. Results

### 2.1. Targeted Integration of Avpr2 and Aqp2 Genes into the Rosa26 Locus in mpkCCDc14 Cells

The *Rosa26* locus on chromosome 6 is frequently exploited for the integration of transgene constructs. To create CRISPR/Cas9-mediated knock-in of rat *Avpr2* and *Aqp2* genes into *Rosa26* locus, we selected sgRNA target sequences (Figure 1A). The sgRosa26, together with a Cas9 mRNA that includes a plasmid coded poly A tail, was selected. Rosa26 homology-directed repair (HDR) donor vector included homology arms (HAs) of 0.8 kb flanking a splicing acceptor (SA)-2A-puromycin element and the V2R-P2A-AQP2 expression cassette (Figure 1A).

To confirm the targeted integration of *Avpr2* and *Aqp2* genes in mpkCCDc14 cells (i.e., V2R-AQP2 cells), junction PCR was performed in two different ways (Figure 1B). The first one was 3′-junction PCR, where the forward primer had homology with integrated gene and the reverse primer had homology with 3′-mpkCCDc14 cells chromosome of *Rosa26* locus (Figure 1A). When the gene was correctly integrated, ~1.3 kb of PCR product could appear (Figure 1B). The second way was 5′-junction PCR, in which its forward primer had homology with 5′-mpkCCDc14 cells chromosome of *Rosa26* locus and reverse primer had homology with integrated gene (Figure 1A). In this case, the PCR product could appear at ~1.5 kb (Figure 1B). 3′- and 5′-Junction PCR revealed that the V2R-AQP2 expression cassette with a long insert size (~2.2 kb) was correctly integrated (Figure 1B).

In addition, the PCR products (V2R and AQP2) from the V2R-AQP2 cells were shown at the predicted sizes, demonstrating that V2R-AQP2 expression cassette was correctly integrated (Figure 1C). PCR experiments in control mpkCCDc14 cells and V2R-AQP2 cells demonstrated that AQP2 and V2R expression was much stronger in the V2R-AQP2 cells (Figure 1C). Immunoblotting revealed that the protein expression of AQP2 (unglycosylated band) was strongly observed in V2R-AQP2 cells in the absence of vasopressin stimulation (Figure 1D). The results indicated that *Avpr2* and *Aqp2* genes were correctly integrated, and the gene product was observed constitutively, even in the absence of vasopressin stimulation.

The engineered V2R-AQP2 cell model was designed to express both the constitutively-expressed AQP2 (integrated) as well as vasopressin-responsive AQP2 endogenously expressed in mpkCCDc14 cells. In particular, we inserted the rat *Aqp2* gene as an integrated *Aqp2* gene to detect two AQP2 forms at the transcript level, not affecting the function of AQP2 proteins. As shown in RT-qPCR experiment (Appendix A), the expression of two distinct forms can be clearly validated using species-specific primers for transcripts.

### 2.2. The Functional Properties of V2R-AQP2 Knock-In mpkCCDc14 Cells

Cell proliferation assay was performed to examine whether the targeted integration of *Avpr2* and *Aqp2* genes in mpkCCDc14 affects cell proliferation (Figure 2A). Cell Counting Kit-8 (CCK-8) analysis showed that the cell proliferation rate of the V2R-AQP2 cells (92 ± 3%) was not different compared to the control mpkCCDc14 cells (100 ± 3%, not significant (n.s.), Figure 2A), indicating that the targeted integration of *Avpr2* and *Aqp2* genes into the *Rosa26* locus did not affect cell proliferation in mpkCCDc14 cells.

Since cAMP/PKA signaling is a main regulatory pathway for vasopressin-mediated AQP2 expression and trafficking in the renal collecting duct, intracellular cAMP levels were measured in control mpkCCDc14 cells and V2R-AQP2 cells (Figure 2B). After the pre-treatment of 3-isobutyl-1-methylxanthine (IBMX, 1 mM) for 15 min, cells were treated with two different concentrations of dDAVP (10^−11^ M or 10^−10^ M) for 15 min in the presence of IBMX. dDAVP (10^−10^ M) treatment significantly increased intracellular cAMP levels in both cell types to a similar level, demonstrating that vasopressin-induced cAMP production was not affected by the targeted integration of *Avpr2* and *Aqp2* genes.

### 2.3. Expression of Functional AQP2 in Control mpkCCDc14 Cells and V2R-AQP2 Cells

AQP2 expression was very low in control mpkCCDc14 cells cultured in vitro system without vasopressin stimulation. We examined whether the targeted integration of *Avpr2* and *Aqp2* genes in mpkCCDc14 cells (V2R-AQP2 cells) enables to exhibit detectable AQP2 protein abundance, even in the absence of vasopressin stimulation. To address the question, two different protocols were used.

In Protocol 1, control mpkCCDc14 cells and V2R-AQP2 cells were treated with dDAVP (10^−9^ M) applied to the basolateral side for 0 h, 3 h, or 6 h after starvation for 24 h (Figure 3A). Despite the absence of dDAVP stimulation (at 0 h), immunoblotting revealed that the abundance of glycosylated AQP2 (~35–50 kDa) was significantly higher in V2R-AQP2 cells, compared with control mpkCCDc14 cells (309 ± 31% of control at 0 h, *p* < 0.05, Figure 3B,C). Moreover, integrated AQP2 protein (bottom of the lower three AQP2 bands) was detected in V2R-AQP2 cells in the absence of dDAVP stimulation (Figure 3B,E), indicating that basal AQP2 protein abundance was more abundant in V2R-AQP2 cells. Short-term dDAVP treatment for 3 h or 6 h further increased both glycosylated and integrated AQP2 abundance in V2R-AQP2 cells (glycosylated AQP2: 532 ± 28% of control at 3 h, *p* < 0.05; 506 ± 15% of control at 6 h, *p* < 0.05, Figure 3B,C and integrated AQP2: 170 ± 5% at 3 h, *p* < 0.05, 168 ± 11% at 6 h, *p* < 0.05, Figure 3B,E), possibly resulted from the prolonged half-life of AQP2 protein. In contrast, the glycosylated AQP2 protein abundance in control mpkCCDc14 cells was not changed after short-term dDAVP stimulation for 3 h or 6 h (Figure 3B,C). A dDAVP treatment for 6 h induced the expression of endogenous AQP2 (middle of the lower three AQP2 bands) to similar level in both cell types (97 ± 6% of control at 6 h, not significant (n.s.), Figure 3B,D).

We performed real-time quantitative PCR (RT-qPCR) using mouse- or rat-specific Aqp2 primers to examine the constitutive expression of integrated rat *Aqp2* gene in V2R-AQP2 cells and to determine the dDAVP-induced upregulation of Aqp2 mRNA. Firstly, to verify the specificity of mouse or rat Aqp2 primer, we prepared mouse kidney IMCD tubule suspension treated with 10^−9^ M dDAVP for 2 h. Mouse Aqp2 primers not only amplified mouse Aqp2 mRNA, but also showed an increase in Aqp2 mRNA expression in response to dDAVP stimulation for 2 h (mouse Aqp2 primer 1: 174 ± 20%, *p* < 0.05; mouse Aqp2 primer 2: 157 ± 49%, not significant (n.s.), Appendix A). In contrast, rat Aqp2 primers did not amplify mouse Aqp2 mRNA in mouse kidney IMCD tubule suspension (Appendix A), indicating the specificity the mouse- or rat-AQP2 primers.

Next, the validated mouse and rat Aqp2 primers were applied to both control mpkCCDc14 cells and V2R-AQP2 cells treated with dDAVP as performed in Protocol 1. RT-qPCR with mouse Aqp2 primer 1 showed that Aqp2 mRNA was amplified (Appendix A) and significantly increased by dDAVP stimulation for 6 h in both cell types (fold change: 89-fold increase in the control mpkCCDc14 cells at 6 h, *p* < 0.05; 79-fold increase in the V2R-AQP2 cells at 6 h, *p* < 0.05, Figure 3F). In contrast, rat Aqp2 primer 1 amplified Aqp2 mRNA only in the V2R-AQP2 cells, in which rat Aqp2 was inserted (Appendix A), but not Aqp2 mRNA in control mpkCCDc14 cells (Appendix A), indicating that rat Aqp2 was inserted as an integrated *Aqp2* gene in V2R-AQP2 cells. Moreover, Aqp2 mRNA expression in V2R-AQP2 cells was increased by dDAVP treatment for 3 h (fold change: 4.86-fold increase at 3 h, *p* < 0.05, Figure 3G). The results indicate that the targeted integration of *Avpr2* and *Aqp2* genes did not affect the dDAVP-induced upregulation of endogenous AQP2 in genome-engineered mpkCCDc14 cells (V2R-AQP2 cells).

In Protocol 2, we examined the effects of dDAVP (10^−9^ M) treatment for 30 min on AQP2 expression and AQP2 phosphorylation at serine 256 (pS256-AQP2) (Figure 3H-O, Appendix A-C). To induce the AQP2 expression, both control mpkCCDc14 cells and V2R-AQP2 cells were treated with dDAVP (10^−9^ M) applied to the basolateral side for 1 day or 2 days (Figure 3H). The dDAVP stimulation was withdrawn for 2 h and then cells were treated again for 30 min (Figure 3H). In dDAVP-nontreated groups (day 0, 30 m−), the glycosylated AQP2 and pS256-AQP2 expression levels (~35–50 kDa) in V2R-AQP2 cells were more abundant, compared with control mpkCCDc14 cells [glycosylated AQP2: 505 ± 110% of control at (day 0, 30 m−), Figure 3I,J; glycosylated pS256-AQP2: 205 ± 21% of control at (day 0, 30 m−), Figure 3I,K]. The integrated AQP2 protein (bottom of the lower three AQP2 bands) was detected in dDAVP-nontreated group of V2R-AQP2 cells (Figure 3I,N), demonstrating the constitutive expression of integrated AQP2 in genome-engineered mpkCCDc14 cells. Moreover, in the groups treated with dDAVP for 30 min without an AQP2 induction period (day 0, 30 m+), the glycosylated AQP2 and pS256-AQP2 were also higher in V2R-AQP2 cells, compared with control mpkCCDc14 cells (glycosylated AQP2: 424 ± 88% of control at (day 0, 30 m+), Figure 3I,J; glycosylated pS256-AQP2: 223 ± 39% of control at (day 0, 30 m+), Figure 3I,K). This finding demonstrated that dDAVP treatment for 30 min was not sufficient to induce endogenous AQP2 expression in control mpkCCDc14 cells. The integrated AQP2 protein was also detected in V2R-AQP2 cells treated with dDAVP for 30 min without an AQP2 induction period (day 0, 30 m+) (Figure 3I,N), which levels were unchanged after dDAVP treatment for 1 day or 2 days. In contrast, when cells were treated with dDAVP for 1 day or 2 days (induction period), the endogenous AQP2 and pS256-AQP2 (middle of the lower three AQP2 bands) was detected (Figure 3I,L,M) and the increase in the expression of glycosylated AQP2 and pS256-AQP2 was comparable in both control mpkCCDc14 cells and V2R-AQP2 cells (Figure 3I–K). The findings indicates that endogenous AQP2 expression was induced by dDAVP in control mpkCCDc14 cells.

### 2.4. Constitutive and Homogenous Expression of the Integrated AQP2 in V2R-AQP2 Cells

Immunocytochemistry of AQP2 was performed to examine the homogeneity of AQP2 expression as well as AQP2 translocation to the plasma membrane in the absence or the presence of dDAVP treatment. dDAVP (10^−9^ M) was applied to the basolateral side for 30 min without a period of dDAVP-mediated AQP2 induction. The relative intensity of AQP2 labeling was more prominent in V2R-AQP2 cells than in control mpkCCDc14 cells in the absence of dDAVP (Figure 4C vs. Figure 4A), indicating the constitutive expression of integrated AQP2 in V2R-AQP2 cells, as demonstrated in Figure 3B. Moreover, AQP2 immunolabeling was more homogenous in V2R-AQP2 cells compared with control mpkCCDc14 cells (Figure 4C vs. Figure 4A), indicating the homogenous expression of integrated AQP2 in V2R-AQP2 cells. Furthermore, prominent AQP2 immunolabeling in the apical domain of the genome-edited cells was observed in response to dDAVP stimulation, similar to that in the unedited control mpkCCDc14 cells (Figure 4D vs. Figure 4B). Consistent with AQP2 immunolabeling, the relative intensity of pS256-AQP2 labeling was more prominent in V2R-AQP2 cells than in control mpkCCDc14 cells in the absence of dDAVP (Figure 4E vs. Figure 4G). dDAVP stimulation induced prominent of pS256-AQP2 immunolabeling in the plasma membrane, similar level in both cell types, as observed in the x-z images (Figure 4F vs. Figure 4H and x-z image). The finding indicated that mpkCCDc14 cells constitutively expressing V2R and AQP2 via genome engineering could be exploited for AQP2 trafficking studies.

## 3. Discussion

In the present study, mpkCCDc14 cells were knocked-in with the rat *Avpr2* gene and *Aqp2* gene (i.e., V2R-AQP2 cells) by exploiting CRISPR/Cas9-mediated genome engineering technology. We demonstrated that the V2R-AQP2 gene cassette with a long insert size (~2.2 kb) was correctly integrated in the *Rosa26* locus on chromosome 6 in mpkCCDc14 cells by 3′- and 5′-junction PCR. Moreover, PCR and immunoblotting revealed the expression of products of integrated *Avpr2* and *Aqp2* genes. Cell proliferation rate and dDAVP-induced cAMP production were not affected by the knock-in of *Avpr2* and *Aqp2* genes in mpkCCDc14 cells. The basal expression level of AQP2 without dDAVP stimulation was significantly higher in V2R-AQP2 cells compared with control mpkCCDc14 cells. Immunocytochemistry demonstrated that V2R-AQP2 cells exhibited more homogenous and prominent AQP2 labeling intensity in the absence of dDAVP stimulation. Moreover, prominent AQP2 immunolabeling (both AQP2 and pS256-AQP2) in the apical domain of the genome-edited cells (x-z images) was observed in response to dDAVP stimulation, similar to that in the unedited control mpkCCDc14 cells.

The mpkCCD cells originally developed by Vandewalle and colleagues were derived from isolated cortical collecting duct (CCD) microdissected from the kidney of SV-PK/Tag transgenic male mice [26]. mpkCCDc14 cells have been used for AQP2 studies, however, their endogenous AQP2 protein abundance is very low without vasopressin stimulation. Moreover, AQP2 expression is not homogenous among cultured cells. Therefore, several limitations lie in the AQP2 study, including trafficking and protein abundance, using mpkCCD cells. Firstly, cells must be treated with vasopressin in advance to induce sufficient AQP2 expression. This makes it difficult to study the shuttling or internalization of AQP2-expressing vesicles in response to vasopressin stimulation or withdrawal. Secondly, there is the heterogeneity of AQP2 protein expression among cells. Thus, quantitative measurement and comparison in the AQP2 protein abundance are not simple. Moreover, although we did not study the expression levels of V2R, V2R expression may also be different among cells, which make the interpretation of vasopressin-induced trafficking or expression of AQP2 difficult. These limitations, therefore, could be overcome by the targeted integration of *Avpr2* and *Aqp2* genes via genome-editing technology in mpkCCDc14 cells.

Several previous attempts were made to obtain the cells highly expressing endogenous AQP2 protein. For instance, the mpkCCDc11 cells were recloned by selecting from the mpkCCDc14 cells expressing different levels of AQP2 protein in the presence of dDAVP [23,24]. The mpkCCDc11 cells were more responsive to dDAVP with changes in AQP2 expression, phosphorylation, apical trafficking, and water permeability. LLC-PK1 cells, an epithelial cell line derived from pig kidney, express the V2R and increase the intracellular cAMP by vasopressin stimulation. Katsura et al. [27] established the LLC-PK1 cell line stably expressing AQP2 (LLC-AQP2 cells). The AQP2 protein in LLC-AQP2 cells was recruited to the plasma membrane in response to dDAVP [28,29] and the intracellular cAMP level was increased by forskolin [27]. Nevertheless, due to the various limitations of the cells listed above (e.g., vasopressin pre-treatment, heterogeneity of AQP2 or V2R expression), we aimed to generate V2R-AQP2 knock-in mpkCCDc14 cells.

In this study, we successfully established V2R-AQP2 knock-in mpkCCDc14 cells using CRISPR/Cas9 genome-engineering technology. The engineered V2R-AQP2 cells expressing both the constitutively-expressed AQP2 (integrated) and vasopressin-responsive AQP2 endogenously expressed in mpkCCDc14 cells could be exploited for AQP2 studies.

## 4. Materials and Methods

### 4.1. Construction of Rosa26 Homology-Directed Repair (HDR) Donor Vector and Knock-In of the V2R-AQP2 Expression Cassette in mpkCCDc14 Cells

Targeting vectors were cloned by using In-Fusion HD Cloning Kit (PT5162-1, Takara, Otsu, Shiga, Japan) according to the manufacturer’s instructions. The backbone vector was pZDonor-AAVS1 Puromycin (D4696, Sigma, St. Louis, MO, USA) having AAVS1 homology arms (left and right) and SA-2A-Puro-Poly A sequences (Figure 1). The CMV promoter from the EGFP vector was inserted upstream of the V2R-P2A-AQP2-pA (Figure 1). Subclone GOI (gene of interest: pCMV-V2R-P2A-AQP2-pA) was inserted into the multiple cloning site (MCS) of the backbone vector. The human codon-optimized Cas9 plasmid, which is also available for a broad spectrum of mammalian cells, including mouse cells (pRGEN-Cas9-CMV, ToolGen, Seoul, Korea), was utilized to produce Cas9 mRNA. The Cas9 plasmid was digested into a linear form with XbaI (R0145S, New England Biolabs, Ipswich, MA, USA) and purified by phenol/chloroform (P2069, Sigma, St. Louis, MO, USA) extraction. 3 M sodium acetate solution (S7899, Sigma, St. Louis, MO, USA) and 100% ethanol (459844, Sigma, St. Louis, MO, USA) were used for precipitating DNA. Capped Cas9 mRNA was produced from a linearized plasmid DNA template by in vitro transcription using MessageMAX T7 ARCA-Capped Message Transcription Kit (MMA60710, Epicentre Biotechnologies, Madison, WI, USA). A poly(A) tail was then added to Cas9 mRNA by polyadenylation with a Poly(A) Polymerase Tailing Kit (PAP5104H, Epicentre Biotechnologies, Madison, WI, USA), which was purified by spin column with MEGAclear Kit (AM1908, Ambion, Austin, TX, USA). *Rosa26*-targeting guide RNA was produced by ToolGen (Seoul, Korea). Briefly, target sequences (5’-ACTCCAGTCTTTCTAGAAGA-3′) were inserted into the pRGEN-U6-sgRNA plasmid. In vitro transcription using MEGAshortscript T7 Transcription Kit (AM1354, Ambion, Austin, TX, USA) was performed to yield the *Rosa26*-guide RNA.

### 4.2. Transfection

To establish V2R-AQP2 knock-in cells (V2R-AQP2 cells), mpkCCDc14 cells were transfected using the Neon electroporation system (MPK5000, Thermo Fisher Scientific, Waltham, MA, USA) according to the manufacturer’s instructions. Cells were washed with DPBS (14190144, Thermo Fisher Scientific, Waltham, MA, USA) and resuspended in Buffer R (Thermo Fisher Scientific, Waltham, MA, USA) at a concentration of 1 × 10^5^ cells/mL. For each electroporation reaction, 1 × 10^5^ cells in 10 μL of Buffer R were mixed with gRNA, Cas9 mRNA, and donor DNA (1:1:1 μg). The mixture was taken up into a 10 μL Neon Pipette Tip and electroporated using the following settings: 1500 V, 30 ms, 1 pulse. Electroporated cells were transferred to a medium supplemented with 10% FBS without antibiotics. After 3–5 days, V2R-AQP2 knock-in mpkCCDc14 cells were selected by treatment with 1 mg/mL of puromycin.

### 4.3. Cell Culture

mpkCCDc14 cells (mouse kidney cortical collecting duct cells) and mpkCCDc14 cells knocked-in with rat *Avpr2* gene and *Aqp2* gene (i.e., V2R-AQP2 cells) were cultured in a 1:1 mixture of DMEM and Ham’s F-12 medium (11330032, Thermo Fisher Scientific, Waltham, MA, USA), containing 60 nM sodium selenite (S9133, Sigma, St. Louis, MO, USA), 5 μg/mL transferrin (T8158, Sigma, St. Louis, MO, USA), 50 nM dexamethasone (D8893, Sigma, St. Louis, MO, USA), 1 nM triiodothyronine (T5516, Sigma, St. Louis, MO, USA), 10 ng/mL epidermal growth factor (E4127, Sigma, St. Louis, MO, USA), 5 μg/mL insulin (I5500, Sigma, St. Louis, MO, USA), 1% antibiotic and antimycotic (15240062, Thermo Fisher Scientific, Waltham, MA, USA), and 2% decomplemented fetal bovine serum (FBS, SV30207.02, Hyclone; Cytiva, Marlborough, MA, USA) at 37 ℃ with 5% CO_2_. For dDAVP treatment, cells were grown in semipermeable filters of the Transwell system (0.4 μm pore size, Transwell Permeable Supports, 3450, Corning, NY, USA).

Protocols for the dDAVP treatment: In Protocol 1 (cell culture without a period of dDAVP-mediated AQP2 induction), cells were cultured for 5 days and starved in serum- and hormone-free media for another 24 h. Then, cells were treated with dDAVP (10^−9^ M, V1005, Sigma, St. Louis, MO, USA), which was applied to the basolateral side of the cells for 3 or 6 h. In Protocol 2 (cell culture having a period of dDAVP-stimulated AQP2 induction), cells were cultured for 6 days in total and lysed. Before lysis, dDAVP (10^−9^ M) was applied to the basolateral side of the cells for 1 day or 2 days. On day 6, dDAVP was withdrawn for 2 h and then cells were treated with dDAVP for 30 min again [24]. The passage numbers of used mpkCCDc14 cells were 29–38. Cells were incubated in a humidified incubator at 37 ℃ with 5% CO_2_.

### 4.4. Genomic DNA Extraction and Junction PCR Analysis of Genomic Integration

Genomic DNA from cultured cells was isolated with DirectPCR Lysis Reagent (Tail) (102-T, Viagen Biotech, Los Angeles, CA, USA). Cells in the 12-well plates were washed with DPBS and suspended in 70 μL of DirectPCR Lysis Reagent containing 0.2 mg/mL Proteinase K (25530049, Thermo Fisher Scientific, Waltham, MA, USA). The samples were incubated for 3–5 h at 55 ℃ until the cells were totally dissolved. To inactivate proteinase K, crude lysates were incubated for 45 min at 85 ℃. For each junction PCR reaction, 5 μL of lysates and AccuPower PCR PreMix (K-2012, Bioneer, Seoul, Korea) were used. Primers used for the detection of HDR-mediated genomic integration had the following sequences: 5′-junction F: CGGAACTCTGCCCTCTAACG, 5′-junction R: TGAGGAAGAGTTCTTGCAGCT, 3′-junction F: CAGTGGCAGCCAGGTTAGC, 3′-junction R: CCTGGGATACCCCGAAGAGT.

### 4.5. Polymerase Chain Reaction (PCR)

The cDNAs were synthesized from 2 μg of total RNA isolated from mpkCCDc14 cells or V2R-AQP2 cells, respectively, using the Prime Script cDNA Synthesis kit (6110A, Takara, Otsu, Shiga, Japan). For DNA amplification, 1 μL of cDNA and AccuPower PCR PreMix (K-2012, Bioneer, Seoul, Korea) were used, according to manufacturer’s instructions. The PCR reaction was performed at 95 ℃ for 3 min followed by 25 cycles at 95 ℃ for 30 s, 55 ℃ for 30 s, and 72 ℃ for 30 s. The primer sequences used for PCR were AQP2-F: CTGCCATCCTCCATGAGATT, AQP2-R: GGAGCAACCGGTGAAATAGA, V2R-F: ACAGCAGCCAGGAGGAACTA, and V2R-R: GCACTTGAAACAGAGCCACA.

### 4.6. Cell Counting Kit (CCK)-8 Assay

Cells were seeded into 96-well culture plates at a density of 5 × 10^4^ cells per well. Following incubation for 24 h at 37 ℃, the CCK-8 assay was performed according to the manufacturer’s instructions. Briefly, 10 μL of CCK-8 solution (CK04, Dojindo Molecular Technologies, Inc., Rockville, MD, USA) was added to each well and the cells were incubated for 2 h at 37 ℃. The absorbance was measured at 450 nm wavelength using a microplate reader.

### 4.7. cAMP Assay

Cells were cultured in a 6-well Transwell system (0.4 μm pore size, Transwell Permeable Supports, 3450, Corning, NY, USA) for 5 days and starved for the next 24 h. All measurements were performed in the presence of 1 mM 3-isobutyl-1-methylxanthine (IBMX; I5879, Sigma, St. Louis, MO, USA) to inhibit cyclic nucleotide phosphodiesterase. After pretreatment of 1 mM IBMX for 15 min, vehicle or dDAVP (10^−11^ M or 10^−10^ M) was added to the basolateral side of the cells for 15 min in the presence of IBMX. Intracellular cAMP levels were measured using a competitive enzyme immunoassay (501040, Cayman Chemical, Ann Arbor, MI, USA) according to the manufacturer’s instructions. The results were presented in picomoles per milliliter of cell lysate.

### 4.8. Semiquantitative Immunoblotting Analysis

The cell lysates were obtained in 1X Laemmli buffer (10 mM Tris-HCl, pH 6.8, 1.5% SDS), including protease and phosphatase inhibitor cocktail (Halt^TM^ Protease and Phosphatase Inhibitor Cocktail 100X, 78440, Thermo Fisher Scientific, Waltham, MA, USA). They were placed in the QIAshredder column (79656, QIAGEN, Hilden, Germany) and centrifuged at 10,000× *g* for 2 min at room temperature. The total protein concentration was measured using the BCA assay kit (Pierce BCA protein assay reagent kit; 23227, Thermo Fisher Scientific, Waltham, MA, USA). Semiquantitative immunoblotting was performed, as previously described [21,22]. Primary antibodies used were anti-AQP2 (1:1000, AB3274, Merk Millipore, Burlington, MA, USA), anti-phosphorylated-AQP2 at serine 256 (1:1000, K0307AP) [30], and anti-β-actin (1:400,000, A1978, Sigma, St. Louis, MO, USA). Immunoblots were visualized by horseradish peroxidase-conjugated secondary antibodies (P447, P448, DAKO, Glostrup, Denmark). Densitometric values were corrected by the densitometry value of β-actin and band density was quantitated by ImageJ (NIH).

### 4.9. IMCD Tubule Suspension

Fresh inner medulla (IM) were obtained from male C57BL/6 mice (7 weeks, Charles River, Orient Bio, Seongnam, Korea) kidneys as previously described [21]. Mice were anesthetized under enflurane inhalation. The kidney IM was dissected, minced, and digested in IMCD suspension buffer (10 mM triethanolamine and 250 mM sucrose, pH 7.6) contatining 3 mg/mL collagenase B (11088807001, Roche, Basel, Switzerland) and 2 mg/mL hyaluronidase (H3884, Sigma, St. Louis, MO, USA). The IMCD suspension was incubated in a warm water bath at 37 ℃ for 90 min. Then, the IMCD tubule suspension was centrifuged at 80× *g* for 30 s to separate the IMCD-enriched fraction (pellet) and non-IMCD fraction (supernatant). The IMCD-enriched fraction was washed twice with ice-cold IMCD suspension buffer containing 0.1% bovine serum albumin (BSA) and treated with 10^−9^ M dDAVP for 2 h. For RNA extraction, the IMCD tubule suspension was homogenized in Trizol using an Ultra-Turrax T10 basic homogenizer (IKA Labortechnik, Staufen, Germany).

### 4.10. Real-Time Quantitative PCR

Total RNA was prepared by Direct-zol^TM^ RNA MiniPrep (R2050, Zymo Research, Irvine, CA, USA) in accordance with the manufacturer’s protocol. cDNAs were synthesized using 500 ng of total RNA by PrimeScript 1st strand cDNA Synthesis Kit (6110A, Takara, Otsu, Shiga, Japan), followed the manufacturer’s instruction. The RT-qPCR was performed using a QuantiTect SYBR Green PCR Kit (204143, QIAGEN, Hilden, Germany) to examine the relative expression of the AQP2 mRNA. β-actin mRNA was used as an internal control and each sample was tested in duplicate. RT-qPCR was run on Rotor-Gene-A (QIAGEN, Hilden, Germany) and threshold was set by 0.02 to determine the threshold cycle (Ct) value. The relative mRNA expression was calculated, as we described previously [21]. The primer sequences used for RT-qPCR were listed in Appendix A.

### 4.11. Immunofluorescence

Cells were grown on semipermeable filter supports in a transwell chamber (0.4 μm pore size, Transwell Permeable Supports, 3460, Corning, NY, USA) and treated with vehicle or 10^−9^ M dDAVP applied to the basolateral side for 30 min without a period of dDAVP-mediated AQP2 induction. Cells were washed twice in PBS and then fixed with 4% paraformaldehyde (P2031, Biosesang, Seoul, Korea) for 30 min at room temperature. Cells were washed twice in PBS and permeabilized with 0.3% Triton X-100 (T8787, Sigma, St. Louis, MO, USA) for 15 min at room temperature. Following permeabilization, cells were incubated with anti-AQP2 antibody (1:200, AB3274, Millipore, Burlington, MA, USA) and anti-phosphorylated AQP2 at serine 256 (1:100, ab111346, Abcam, Cambridge, UK) in PBS at 4 ℃ overnight. Cells were washed and incubated with goat-anti-rabbit IgG Alexa Fluor 488 secondary antibody (A11008, Invitrogen, Carlsbad, CA, USA) for 2 h at room temperature. The nuclei were stained with DAPI (D1306, Invitrogen, Carlsbad, CA, USA) for 30 min at room temperature and cells were mounted with an antifading reagent (P36934, Invitrogen, Carlsbad, CA, USA). Immunofluorescence microscopy was carried out using a laser scanning confocal microscope (Zeiss LSM 5 EXCITER, Jena, Germany).

### 4.12. Stastical Analysis

Quantitative data were presented as means ± SEM. Comparisons between two groups were made by the unpaired *t*-test. A comparison of multiple groups was made by one-way ANOVA followed by Tukey’s multiple comparisons test. Multiple comparison tests were only applied when a significant difference was determined in ANOVA. GraphPad Prism software (GraphPad Software, La Jolla, CA, USA) was used for all statistical analysis. *p* values of <0.05 were considered statistically significant.

## Figures and Tables

**Figure 1 ijms-24-01684-f001:**
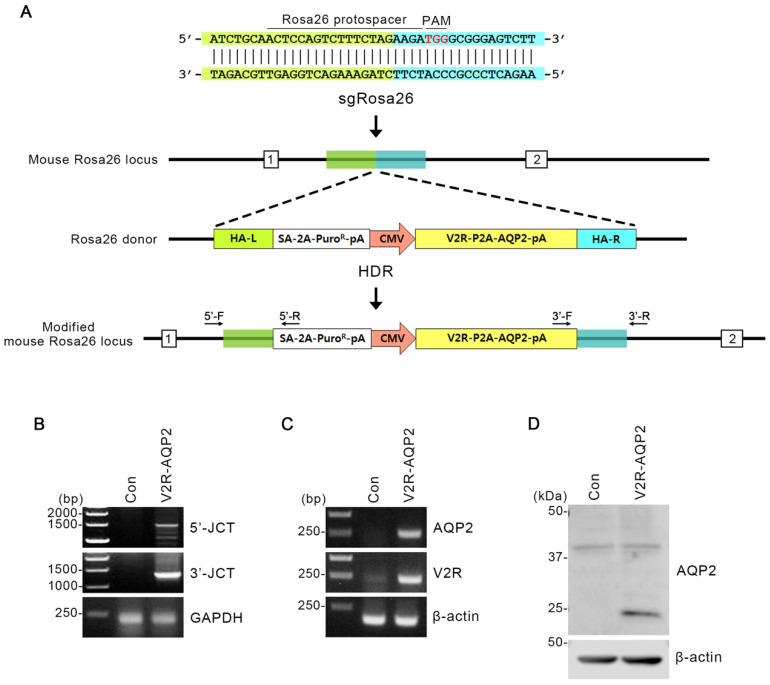
Integration of V2R-AQP2 expression cassette into the *Rosa26* locus using CRISPR/Cas9 genome-engineering technology in mpkCCDc14 cells. (**A**) Schematics of integration into the *Rosa26* locus in mpkCCDc14 cells. Protospacer targeting the *Rosa26* locus and the PAM element were indicated. The Rosa26 homology-directed repair (HDR) donor vector included homology arms (HAs) of 0.8 kb flanking a SA-2A-puromycin element and the V2R-P2A-AQP2 expression cassette. HA-left (L) was indicated in green and HA-right (R) was in blue. Two different junction PCR sites (3′- and 5′-junction) were depicted. (**B**) Junction PCR analysis in V2R-AQP2 cells to identify the targeted integration of *Avpr2* and *Aqp2* genes. PCR with 5′-F/5′-R and 3′-F/3′-R primers amplified the 5′-junction PCR product (1477 bp) and 3′-junction PCR product (1271 bp) of the V2R-AQP2 cells. (**C**) PCR showing the expression of AQP2 and V2R in control mpkCCDc14 cells and V2R-AQP2 cells. The predicted PCR product size of AQP2 and V2R were 245 bp and 216 bp, respectively. (**D**) Immunoblotting of AQP2 in control mpkCCDc14 cells and V2R-AQP2 cells. PAM, protospacer adjacent motif; HA-L, homologous arm-left; HA-R, homologous arm-right; pA, poly A; Puro, puromycin resistance gene; CMV, cytomegalovirus promoter; HDR, homology-directed repair; 5′-JCT, 5′-junction; 3′-JCT, 3′-junction; Con, control mpkCCDc14 cells; V2R-AQP2, mpkCCDc14 cells constitutively expressing V2R and AQP2 via CRISPR/Cas9-mediated genome engineering technology.

**Figure 2 ijms-24-01684-f002:**
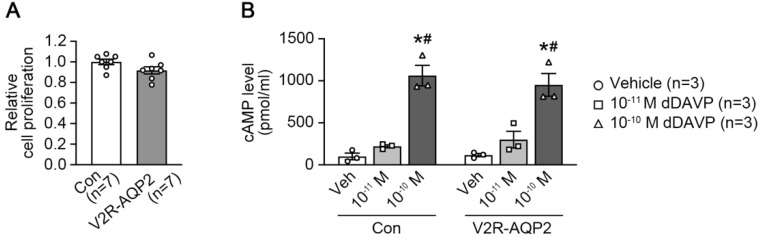
Cell proliferation and dDAVP-induced intracellular cAMP levels in V2R-AQP2 cells. (**A**) Cell proliferation assay (CCK-8) of control mpkCCDc14 cells and V2R-AQP2 cells. *n* = 7 in each group. (**B**) Intracellular cAMP levels were analyzed in control mpkCCDc14 cells and V2R-AQP2 cells. Following pre-incubation of 1 mM IBMX for 15 min, cells were treated with vehicle, 10^−11^ M dDAVP or 10^−10^ M dDAVP for 15 min in the continued presence of IBMX. *n* = 3 in each group. * *p* < 0.05 compared with vehicle group in control mpkCCDc14 cells or V2R-AQP2 cells, respectively, ^#^
*p* < 0.05 compared with 10^−11^ M dDAVP group in control mpkCCDc14 cells or V2R-AQP2 cells, respectively.

**Figure 3 ijms-24-01684-f003:**
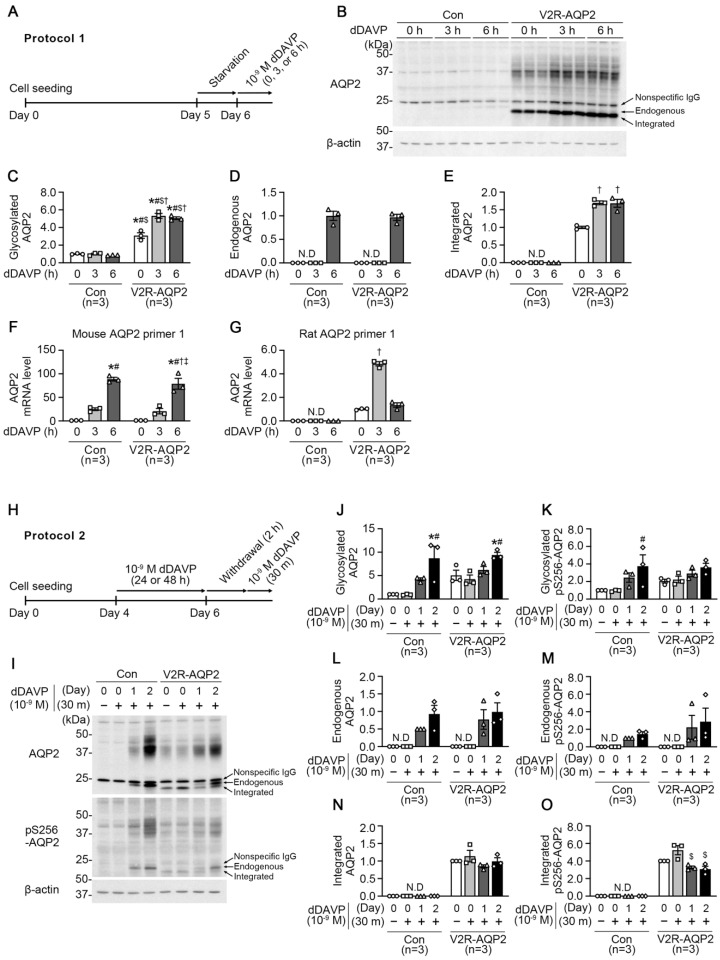
Semiquantitative immunoblotting and real-time quantitative PCR of AQP2 and phosphorylated AQP2 at serine 256 (pS256-AQP2) in control mpkCCDc14 cells and V2R-AQP2 cells. (**A**) Experimental design for dDAVP stimulation (Protocol 1, Panels **A**–**G**). After starvation for 24 h, 10^−9^ M dDAVP was applied to the basolateral side of the cells in the transwell chamber for 0 h, 3 h, or 6 h. (**B**–**E**) Semiquantitative immunoblotting of AQP2 in total cell lysates of control mpkCCDc14 cells and V2R-AQP2 cells treated with dDAVP as in Protocol 1. *n* = 3 in each group. (**F**,**G**) Real-time quantitative PCR with primers specific for mouse- or rat-Aqp2 mRNA in control mpkCCDc14 cells and V2R-AQP2 cells treated with dDAVP as in Protocol 1. *n* = 3 in each group. * *p* < 0.05 compared with 0 h of control mpkCCDc14 cells. ^#^
*p* < 0.05 compared with 3 h of control mpkCCDc14 cells. ^$^
*p* < 0.05 compared with 6 h of control mpkCCDc14 cells. ^†^
*p* < 0.05 compared with 0 h of V2R-AQP2 cells. ^‡^
*p* < 0.05 compared with 3 h of V2R-AQP2 cells. (**H**) Experimental design for dDAVP stimulation (Protocol 2, Panels **H**–**O**). For induction of AQP2 expression, cells were treated with 10^−9^ M dDAVP to the basolateral side for 1 day or 2 days. After that, dDAVP was withdrawn for 2 h and then cells were treated again for 30 min. (**I**–**O**) Immunoblotting of AQP2 and pS256-AQP2 in control mpkCCDc14 cells and V2R-AQP2 cells treated with dDAVP as in Protocol 2. * *p* < 0.05 compared with control mpkCCDc14 cells (day 0, 30 m−). ^#^
*p* < 0.05 compared with control mpkCCDc14 cells (day 0, 30 m+). ^$^
*p* < 0.05 compared with V2R-AQP2 cells (day 0, 30 m−).

**Figure 4 ijms-24-01684-f004:**
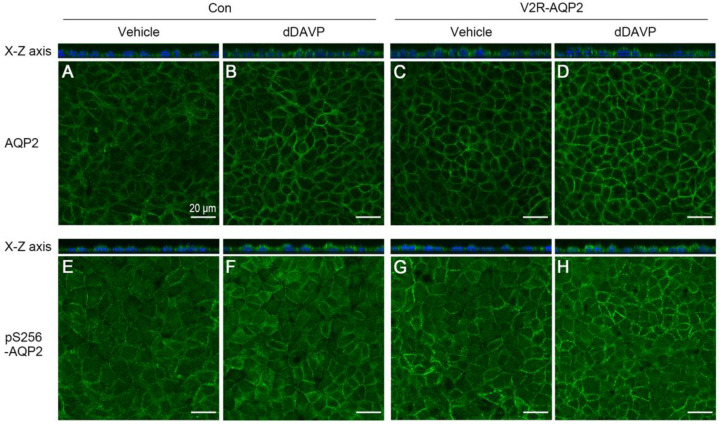
Immunofluorescence of total AQP2 and pS256-AQP2 in control mpkCCDc14 cells and V2R-AQP2 cells. Immunofluorescence staining of total AQP2 (**A**–**D**) and pS256-AQP2 (**E**–**H**) in control mpkCCDc14 cells and V2R-AQP2 cells treated with dDAVP (10^−9^ M) to the basolateral side for 30 min without an AQP2 induction period. Con, control mpkCCDc14 cells; V2R-AQP2, mpkCCDc14 cells constitutively expressing V2R and AQP2 via CRISPR/Cas9-mediated genome engineering technology. Scale bar is 20 μm.

## Data Availability

Not applicable.

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
