# Peer review of "Genome-Engineered mpkCCDc14 Cells as a New Resource for Studying AQP2"

_ijms, 2023, doi:10.3390/ijms24021684_

Round 1

Reviewer 1 Report

The authors aim to generate an mpkCCDc14 cell line which stably expresses V2R and AQP2 in order to allow studies of Aquaporin 2 trafficking. The main concern for this paper is that the data provided are unconvincing and that no application of the edited cells is provided. In order to support the claim that V2R-AQP2 cells offer an advantage with respect to unedited cells, substantial evidence, in the form of experiments demonstrating the use of V2R-AQP2 cells for Aquaporin 2 trafficking, must be provided. These could even be applied to studies of Sorting nexin 27 (SNX27), a protein already investigated by the authors. In its current form, the article is incomplete and must provide applications for the V2R-AQP2 cells.

Other major concerns:

-        The authors used CRISPR-Cas9 technology to insert a gene cassette containing rat Avpr2 and Aqp2 yet provide no explanation for why mouse Avpr2 and Aqp2 were not used. It seems that there may be different patterns of Aquaporin 2 glycosylation between rat and mouse Aquaporin 2 since the major differences between normal and edited cells is in non-glycosylated Aquaporin 2.

-        The antibody that the authors have used, Rabbit Aquaporin 2, Millipore AB3274, appears to be specific for the human and rat isoforms, however it also seems to detect the mouse isoform in unedited cells. Have the authors validated this antibody for mouse Aquaporin 2.

-        The western blots shown in Figure 3 for Aquaporin appear to produce at least two bands. The upper one (approx. 40 kDa) may represent glycosylated Aquaporin 2 while the lower one (approx. 25 kDa) may represent the non-glycosylated form. This must be explained either in the text or in the figure, as it is not clear which bands the authors refer to. Additionally, when densitometry is performed it is unclear which bands were quantified.

-        The authors use Western Blotting to detect expression of Aquaporin 2, then use densitometry to perform statistical tests. I would suggest the use of RT-PCR and ELISA to more convincingly determine the upregulation of Aquaporin 2. Additionally, I would recommend that the authors also verify the levels of V2R and compare the levels between normal and edited cells.

-        In Fig.3C the authors have 3 replicates for each condition, while in Fig.3D it seems that only a representative blot is shown. It is unclear how many experimental replicates were carried out for the data in Fig.3D and this must be stated either in the results section or the figure legend.

-        In Fig.3D there is no evaluation of Aquaporin 2 expression after 1 and 2 days of dDAVP pre-treatment without 30 mins of dDAVP administration. This is an important control that is missing and could help determine whether V2R-AQP2 cells indeed express more Aquaporin 2 than unedited cells.

-        The authors do not address or provide an explanation for the fact that unedited cells express visibly more glycosylated Aquaporin 2 and pS256-Aquaporin 2 than V2R-AQP2 cells after 1 and 2 days of dDAVP pre-treatment and a 30 mins stimulation with dDAVP. As mentioned above, RT-PCR and ELISA would clarify the degree of upregulation and activation of Aquaporin 2. Along these lines, based on the data the authors provide one may conclude that unedited cells are more suitable for functional studies. The authors must address the two Aquaporin 2 bands on the blot and clarify which ones are of interest. Glycosylation of Aquaporin 2 is important for its translocation to the apical membrane, and hence the glycosylated band may indicate the functional protein.

-        In Fig.3E the authors perform densitometry analysis based on data in Fig.3C, yet they do not carry out the same analysis for the data in Fig.3D. I would suggest densitometric analysis of all blots.

-        Immunofluorescence experiments do not include a condition with dDAVP pre-treatment for 1 and 2 days, which based on data in Fig.3D would appear to be highly relevant for determining the value of V2R-AQP2 cells over unedited cells. This missing condition is very important and must be included in the study. Additionally, in Fig.4 the only apparent difference between the two cell lines appears to be in terms of intensity, not distribution. Furthermore, Immunofluorescence for pS256-Aquaporin 2 would be very important for the comparison between the cell lines. The authors should also perform immunofluorescence for V2R to evaluate whether its expression is increased in V2R-AQP2 cells.

-        The statement by the authors that there is “more prominent AQP2 labelling in the apical plasma membrane after dDAVP stimulation” is not supported by the data in Fig.4. In order to support this claim, the authors must co-localise Aquaporin 2 and a marker for the apical membrane (such as biotin).

Minor concerns:

-        At the start of section 2.3 Fig.3C is described before Fig.3A and Fig.3B. This is confusing and the order in the text must reflect the order of appearance in the figure. The authors could either describe Fig.3A first and then describe the blot (renaming it Fig.3B) or provide the data regarding low Aquaporin 2 in unedited cells in a separate panel from the rest of Fig.3C.

-        In the discussion the second paragraph is redundant since the Rosa26 locus is well known and characterised and there is no need to describe it. Along the same lines, there is no need to explain why CRISPR was used as opposed to ZFNs and TALENs. I would suggest leaving out the entire paragraph.

-        In the discussion in the fourth paragraph the authors discuss other cell lines used for studying Aquaporin 2 trafficking then make a brief comment about how they are not suitable for such purposes. In the past the authors have used mpkCCD cells (Choi, H. J., Jang, H. J., Park, E., Tingskov, S. J., Nørregaard, R., Jung, H. J., & Kwon, T. H. (2020). Sorting nexin 27 regulates the lysosomal degradation of aquaporin-2 protein in the kidney collecting duct. Cells9(5), 1208.) and IMCD cells (Tingskov, S. J., Choi, H. J., Holst, M. R., Hu, S., Li, C., Wang, W., ... & Nørregaard, R. (2019). Vasopressin-independent regulation of aquaporin-2 by tamoxifen in kidney collecting ducts. Frontiers in physiology10, 948.) to study Aquaporin 2. The authors must clarify why those cell lines present limitations and why dDAVP pre-treatment is limiting in performing experiments of Aquaporin 2 trafficking.

-        In Materials and Methods Section 4.1 the authors describe the use of a human codon optimised Cas9 plasmid, however they transfect this plasmid into mouse cells. The authors must explain this choice in the text.

-        In Materials and Methods the section 4.3 Transfection should be described prior to section 4.2 Cell culture.In section 4.2 Cell culture the culture method of V2R-AQP2 cells is described before they are even generated.

-        In Materials and Methods the anti-AQP2 antibody from Millipore has a different catalogue number between Sections 4.8 and 4.9. It would appear that the correct antibody is AB3274; only the correct number should be provided.

-        12 out of the 41 references are self-citations. Many of them could be omitted while conveying the same ideas.

Reviewer 2 Report

Dear authors,

Cells that stably express AQP2 are necessary for AQP2-trafficking research. Therefore, this is an interesting study.

Comments

1. What do the authors think about the significance of the integration of Avpr2 in addition to Aqp2? Why does the abundance of AQP2 protein increase in V2R-AQP2 cells? Because cAMP was elevated in control cells as much as in V2R-AQP2 cells in Figure 2B.

2. We see some sharp lower-molecular bands in addition to the expected band around 25 kDa in the third lane to the eighth lane detected by AQP2 antibody in Figure 3D. What do the authors think about these lower-molecular bands?

3. It should be better to indicate the primer position in Figure 1A.

Thank you very much.

Author Response

Jan 2, 2023

MS#: ijms-1955144_Revised

Dear Editor,

Thank you very much for your evaluation of our paper. According to the reviewers’ comments, we did new experiments and revised the manuscript and figures. We very much hope that you will find the revised manuscript acceptable for publication in the IJMS.

Sincerely,

Tae-Hwan 

Tae-Hwan Kwon, MD, PhD

Department of Biochemistry and Cell Biology

School of Medicine, Kyungpook National University

Dongin-dong 101, Taegu 700-422, South Korea

REVIEWER 2

Comments and Suggestions for Authors

Dear authors,

Cells that stably express AQP2 are necessary for AQP2-trafficking research. Therefore, this is an interesting study.

Comments

  1. What do the authors think about the significance of the integration of Avpr2 in addition to Aqp2? Why does the abundance of AQP2 protein increase in V2R-AQP2 cells? Because cAMP was elevated in control cells as much as in V2R-AQP2 cells in Figure 2B.

The purpose of the study is to develop a cell model with homogenous expression of V2R (Avpr2 gene) and AQP2 (Aqp2 gene) from mpkCCDc14 cells for in vitro studies on vasopressin-responsive AQP2 regulation. Regarding to homogenous AQP2 expression across cultured cells, V2R as a receptor of vasopressin is also a critical component in vasopressin-responsive regulation of AQP2. To avoid misinterpretation that might be caused by heterogenous V2R expression, we integrated both Avpr2 and Aqp2 genes. While unedited cells have only the endogenous Aqp2 gene, V2R-AQP2 cells contain two types of Aqp2 genes, i.e., the endogenous Aqp2 gene and the integrated Aqp2 gene. The insert for the integrated Aqp2 gene does not include vasopressin- or cAMP-responsive promoters. Therefore, V2R-AQP2 cells should have more AQP2 proteins produced from the inserted Aqp2 gene. We now indicated bands on the blot image to clarify AQP2 forms. The upper bands at 37-50 kDa represent glycosylated AQP2. The lower three bands represent integrated AQP2 proteins (bottom), endogenous AQP2 proteins (middle), and IgG (top) (Figure 3B and 3I, Figure S3).

  1. We see some sharp lower-molecular bands in addition to the expected band around 25 kDa in the third lane to the eighth lane detected by AQP2 antibody in Figure 3D. What do the authors think about these lower-molecular bands?

The results from western blot analysis showed both endogenous and inserted AQP2 with different molecular weights and expression patterns under dDAVP treatment. We now indicated bands on the blot image to clarify AQP2 forms (Figure 3).

  1. It should be better to indicate the primer position in Figure 1A.

Thank you for suggestion. We added primer positions in Figure 1A.

Round 2

Reviewer 1 Report

The authors have responded to the reviewers’ comments adequately and addressed the major concerns with the submitted manuscript. The submitted article in its current form presents a sound scientific idea which is explained thoroughly. The figures clearly represent the results and are explained in depth.